# Genome-wide association analyses of chronotype in 697,828 individuals provides insights into circadian rhythms

Samuel E. Jones [1], Jacqueline M. Lane[2,3,4], Andrew R. Wood[1], Vincent T. van Hees [5], Jessica Tyrrell[1], Robin N. Beaumont [1], Aaron R. Jeffries [1], Hassan S. Dashti [2,4], Melvyn Hillsdon[6], Katherine S. Ruth [1], Marcus A. Tuke [1], Hanieh Yaghootkar[1], Seth A. Sharp[1], Yingjie Jie[1], William D. Thompson[1], Jamie W. Harrison [1], Amy Dawes[1], Enda M. Byrne [7], Henning Tiemeier[8,9], Karla V. Allebrandt[10], Jack Bowden[11,12], David W. Ray [13,14], Rachel M. Freathy[1], Anna Murray[1], Diego R. Mazzotti[15], Philip R. Gehrman[16], 23andMe Research Team, Debbie A. Lawlor [11,12], Timothy M. Frayling[1], Martin K. Rutter[13,14,17], David A. Hinds [18], Richa Saxena [2,3,19] & Michael N. Weedon[1]

Being a morning person is a behavioural indicator of a person's underlying circadian rhythm. Using genome-wide data from 697,828 UK Biobank and 23andMe participants we increase the number of genetic loci associated with being a morning person from 24 to 351. Using data from 85,760 individuals with activity-monitor derived measures of sleep timing we find that the chronotype loci associate with sleep timing: the mean sleep timing of the 5% of individuals carrying the most morningness alleles is 25 min earlier than the 5% carrying the fewest. The loci are enriched for genes involved in circadian regulation, cAMP, glutamate and insulin signalling pathways, and those expressed in the retina, hindbrain, hypothalamus, and pituitary. Using Mendelian Randomisation, we show that being a morning person is causally associated with better mental health but does not affect BMI or risk of Type 2 diabetes. This study offers insights into circadian biology and its links to disease in humans.

---

[1] Genetics of Complex Traits, University of Exeter Medical School, Royal Devon & Exeter Hospital, Exeter, EX2 5DW, UK. [2] Center for Genomic Medicine, Massachusetts General Hospital, Boston, 02114 MA, USA. [3] Department of Anesthesia, Critical Care and Pain Medicine, Massachusetts General Hospital and Harvard Medical School, Boston, 02114 MA, USA. [4] Broad Institute, Cambridge, 02142 MA, USA. [5] Netherlands eScience Center, Amsterdam, 1098 XG, Netherlands. [6] Sport and Health Sciences, College of Life and Environmental Sciences, University of Exeter, Exeter, EX1 2LU, UK. [7] The University of Queensland, Institute for Molecular Bioscience, Brisbane, 4072 QLD, Australia. [8] Department of Epidemiology, Erasmus Medical Center, Rotterdam, 3015 GE, Netherlands. [9] Department of Psychiatry, Erasmus Medical Center, Rotterdam, 3015 GD, Netherlands. [10] Department of Translational Informatics, Translational Medicine Early Development, Sanofi-Aventis Deutschland GmbH, Industriepark Höchst, Frankfurt, 65926, Germany. [11] MRC Integrative Epidemiology Unit at the University of Bristol, Bristol, BS8 2BN, UK. [12] Population Health Sciences, Bristol Medical School, University of Bristol, Bristol, BS8 2BN, UK. [13] Faculty of Biology, Medicine and Health, University of Manchester, Manchester, M13 9PL, UK. [14] Division of Endocrinology, Diabetes & Gastroenterology, School of Medical Sciences, Faculty of Biology, Medicine and Health, University of Manchester, Manchester, M13 9PL, UK. [15] Center for Sleep and Circadian Neurobiology, University of Pennsylvania, Philadelphia, 19104 PA, USA. [16] Perelman School of Medicine of the University of Pennsylvania, Philadelphia, 19104 PA, USA. [17] Manchester Diabetes Centre, Manchester University NHS Foundation Trust, Manchester Academic Health Science Centre, Manchester, M13 0JE, UK. [18] 23andMe Inc., 899W. Evelyn Avenue, Mountain View, CA, 94041, USA. [19] Departments of Medicine, Brigham and Women's Hospital and Beth Israel Deaconess Medical Center, Harvard Medical School, Boston, 02115, USA. These authors contributed equally: Samuel E. Jones, Jacqueline M. Lane, Andrew R. Wood. These authors jointly supervised this work: Debbie A. Lawlor, Timothy M. Frayling, Martin K. Rutter, David A. Hinds, Richa Saxena, Michael N. Weedon. A full list of consortium members appears at the end of this paper. Correspondence and requests for materials should be addressed to M.N.W. (email: M.N.Weedon@exeter.ac.uk)

Circadian rhythms are fundamental cyclical processes that occur in most living organisms, including humans. These daily cycles affect a wide range of molecular and behavioural processes, such as hormone levels, core body temperature and sleep–wake patterns[1]. Chronotype, often referred to as circadian preference, describes an individual's proclivity for earlier or later sleep timing and is a physical and behavioural manifestation of the coupling between internal circadian cycles and the need for sleep, driven by sleep homoeostasis. Significant natural variation exists amongst the human population with chronotype typically measured on a continuous scale[2], though individuals are often separated into morning people (or "larks") who prefer going to bed and waking earlier, evening people (or "owls") who prefer a later bedtime and later rising time, and intermediates who lie between the two extremes[3,4]. Age and gender as well as environmental light levels explain a substantial proportion of variation in chronotype, but genetic variation is also an important contributor[5–8].

There is evidence that alterations to circadian timing are linked to disease development, particularly metabolic and psychiatric disorders[9,10]. Animal model studies have shown that mutations in, and altered expression of, key circadian rhythm genes can cause obesity, hyperglycaemia and defective beta-cell function leading to diabetes[11–13]. In humans, there are many reported associations between disrupted circadian rhythms and disease[14,15], but the evidence for a causal role of chronotype on disease is limited[16]. For example, evening people have an increased frequency of obesity[17], Type 2 diabetes[18] and depression[19] independent of sleep disturbance, and studies of shift workers show an increased risk of diabetes, depression and other diseases[20]. However, these associations could be explained by reverse causality (diseases affecting sleep patterns or dictating job options) or confounding (common risk factors influencing both chronotype and disease). Genetic analyses identifying variants robustly associated with putative risk factors, such as chronotype, can improve causal understanding by providing genetic instruments for use in Mendelian Randomisation (MR) analyses[21–23], which minimise the effect of both reverse causality and bias caused by confounding. Identifying genetic variants associated with chronotype and sleep timing will also provide insights into the biological processes underlying circadian rhythms and sleep homoeostasis.

Three previous genome-wide association studies (GWAS)[24–26], using a maximum of 128,286 individuals, identified a total of 24 independent variants associated with self-report chronotype. In this study, we perform a GWAS meta-analysis of a substantially expanded set of 697,828 individuals, including 248,098 participants from 23andMe Inc., a personal genetics company, and 449,734 participants from UK Biobank[27,28]. In addition to confirming an enrichment of circadian rhythm and brain expressed genes at chronotype-associated loci and genetic correlation with mental health disorders[25,26], we identify 327 additional chronotype-associated loci and demonstrate that the chronotype-associated variants are associated with objective measures of sleep timing, but not sleep duration or quality, in 85,760 UK Biobank participants. By fine-mapping the genetic associations at all loci, we identify 10 coding variants with a high likelihood of being the causal variant, providing prospective targets for chronobiological investigation and go on to show evidence of a causal link between chronotype and mental health by MR.

## Results

**Meta-analysis identifies 351 loci associated with chronotype.** We performed a GWAS of self-report chronotype (phenotype summarised in Table 1) using 11,977,111 imputed variants in 449,734 individuals of European-ancestry from the UK Biobank and meta-analysed with summary statistics from a self-report morningness GWAS using 11,947,421 variants in 248,098 European-ancestry 23andMe research participants. We identified 351 independent loci at $P < 5 \times 10^{-8}$, of which 258 reached $P < 6 \times 10^{-9}$, a correction for the significance threshold based on permutation testing (Supplementary Methods). Of the 351 loci, 24 had been previously reported in earlier GWAS of chronotype[24–26] and 327 were novel associations. The primary meta-analysis, based on sample size, and individual study results are shown in Fig. 1 and Supplementary Data 1. Conditional analysis identified 49 loci with multiple independent signals (Supplementary Data 2). A sensitivity analysis was performed in the UK Biobank data alone, excluding shift workers and those either on medication or with disorders affecting sleep (see the Methods section and Supplementary Methods for details). Effect sizes were similar to those in the full UK Biobank GWAS (Supplementary Data 1 and Supplementary Figure 1).

**Known circadian genes amongst associated loci.** Well-documented circadian rhythm genes were among the most strongly associated loci (Supplementary Data 1). These genes included the previously reported loci containing *RGS16*, *PER2*, *PER3*, *PIGK/AK5*, *INADL*, *FBXL3*, *HCRTR2* and *HTR6*[24–26], and newly associated loci containing known circadian rhythm genes *PER1*, *CRY1* and *ARNTL* (Supplementary Figure 2). At the *PER3* locus, two highly correlated low-frequency missense variants (rs150812083 and rs139315125, minor allele frequency (MAF) = 0.5%), previously reported to be a monogenic cause of familial advanced sleep phase syndrome[29], were associated with self-reported morningness (odds ratio (OR) = 1.44 for minor allele; $P = 2 \times 10^{-38}$) but with a lower magnitude of effect on sleep timing than expected in the activity-monitor derived measures of chronotype, advancing sleep timing (as measured by time of minimum activity) by only 8 min (95% confidence interval (CI): 4–13, $P = 4.3 \times 10^{-4}$) as opposed to the average 4.2 h reported in the previous study[29].

**Chronotype loci affect sleep timing but not quality or duration.** Self-report assessments of sleep and chronotype can be subject to reporting bias[30–33]. To assess and quantify the effect of the genetic associations on objective measures of sleep timing, duration and quality, we tested the association of the chronotype-associated variants with sleep estimates derived from the UK Biobank activity monitor data. Derived phenotypes included sleep timing, efficiency and duration. Timing was determined by timings of midpoint of sleep, the least active 5 h of the day (L5 timing) and midpoint of the most-active 10 h of the day (M10 timing). Summary statistics of these derived phenotypes and their associations with self-report morningness are presented in Supplementary Table 1, and their associations with the newly identified chronotype single nucleotide polymorphisms (SNPs) are provided in Supplementary Data 3. To avoid inflation of associations due to overlapping samples, we performed an additional GWAS meta-analysis of self-reported morningness excluding all UK Biobank individuals with activity monitor data. Of the 292 lead chronotype variants reaching $P < 5 \times 10^{-8}$ from this meta-analysis that were available in the UK Biobank imputed genotype data, 258 had a consistent direction of effect for sleep midpoint (binomial test $P = 3.8 \times 10^{-44}$), 262 with L5 timing (binomial $P = 9.3 \times 10^{-48}$) and 260 with M10 timing (binomial $P = 6.4 \times 10^{-46}$). A genetic risk score (GRS) of these 292 variants was associated with earlier sleep midpoint, L5 timing and M10 timing (binomial $P = 4 \times 10^{-128}$, $P = 1 \times 10^{-182}$ and $P = 7 \times 10^{-130}$, respectively). There was little evidence of association between the chronotype GRS and the activity monitor sleep phenotypes that estimate sleep duration and fragmentation (Supplementary Table 2), indicating a specific effect

**Table 1 Distribution and demographics of chronotype in the UK Biobank**

| Chronotype category | Phenotype coding | N | Sex (% male) | Age (SD) | TDI | BMI (S.D.) |
|---|---|---|---|---|---|---|
| Definitely morning | 2 | 107,555 | 43.6 | 57.7 (7.7) | −1.4 | 27.5 (4.8) |
| More morning than evening | 1 | 144,731 | 43.9 | 57.0 (7.9) | −1.7 | 27.1 (4.6) |
| Don't know | 0 | 46,538 | 57.1 | 56.8 (8.0) | −1.43 | 27.3 (4.7) |
| More evening than morning | −1 | 115,090 | 45 | 56.1 (8.2) | −1.41 | 27.4 (4.8) |
| Definitely evening | −2 | 35,818 | 46.8 | 55.3 (8.3) | −1.05 | 27.9 (5.2) |
| All | | 449,732 | 45.7 | 56.8 (8.0) | −1.47 | 27.4 (4.8) |

Summary of sex, age, townsend deprivation index (TDI) and BMI by chronotype categories in European-ancestry individuals from the UK Biobank study. SD denotes standard deviation

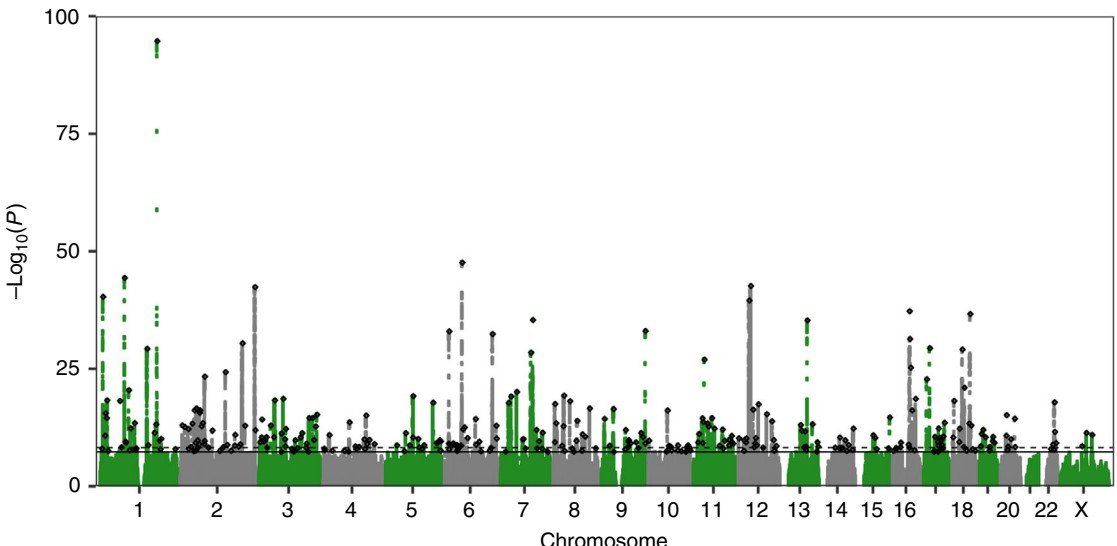

**Fig. 1** Manhattan plot of the chronotype meta-analysis GWAS. The solid line indicates the typical genome-wide significance threshold of $P = 5 \times 10^{-8}$ and the dashed line marks the threshold of $P = 6 \times 10^{-9}$ identified through permutation testing. Lead variants are annotated with a diamond

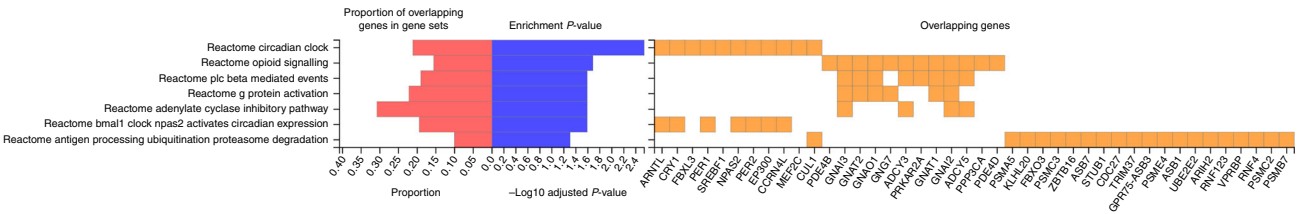

**Fig. 2** Reactome gene sets overlapping Chronotype genes. Chronotype genes were identified using positional and eQTL mapping in FUMA's GENE2FUNC process. Note that these results may differ to those produced by MAGMA

of the chronotype SNPs on sleep timing and circadian metrics. Limiting the analysis to the 109 lead variants identified from the independent 23andMe GWAS gave similar results (Supplementary Table 2). Using the activity-monitor derived estimates of sleeping timing, the 5% of individuals carrying the most morningness alleles at the 292 associated loci had L5 timing shifted earlier, on average, by 25.1 min (95% CI: 22.5–27.6) compared to the 5% carrying the fewest morningness alleles: a mean L5 time of 03:06 rather than 03:32. The data suggest that variants associated with self-report chronotype strongly relate to an individual's sleep timing and therefore represent valid instruments for MR.

**Loci enriched in circadian rhythm pathways and brain tissues.** To identify biological pathways and tissues enriched for genes at the associated loci, we used MAGMA[34], implemented as part of

the FUMA GWAS[35] platform (Figs. 2 and 3, Supplementary Data 4 and Supplementary Table 3). Because of the variety of methods available and databases employed, and to allow better comparisons with studies that have implemented other methods, we also performed secondary gene-set and tissue enrichment using the software packages PASCAL[36], MAGENTA[37] and DEPICT[38] (Supplementary Datas 5–7). We identified strong enrichment in circadian rhythm and circadian clock pathways as with previous morningness GWAS[24–26]. We also identified multiple pathways that correspond to (central) nervous system and brain development, components of neuronal cells such as synapses, axons and dendrites, as well as neurogenesis. There was clear enrichment in all types of brain tissue (Fig. 4, Supplementary Table 3 and Supplementary Data 8), in behavioural pathways, containing genes responsible for mediating behavioural responses to internal and external stimuli, and in retinal tissue

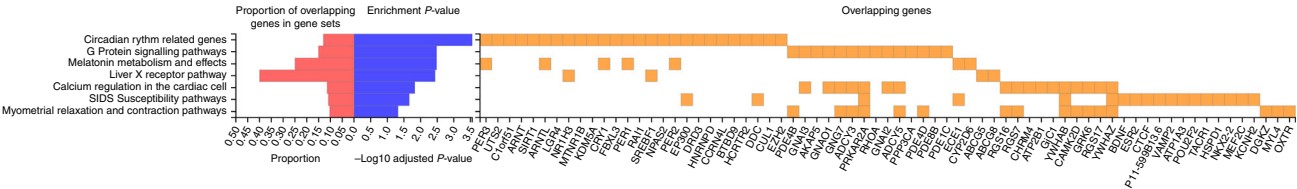

**Fig. 3** WikiPathways gene sets overlapping Chronotype genes. Chronotype genes were identified using positional and eQTL mapping in FUMA's GENE2FUNC process. Note that these results may differ to those produced by MAGMA

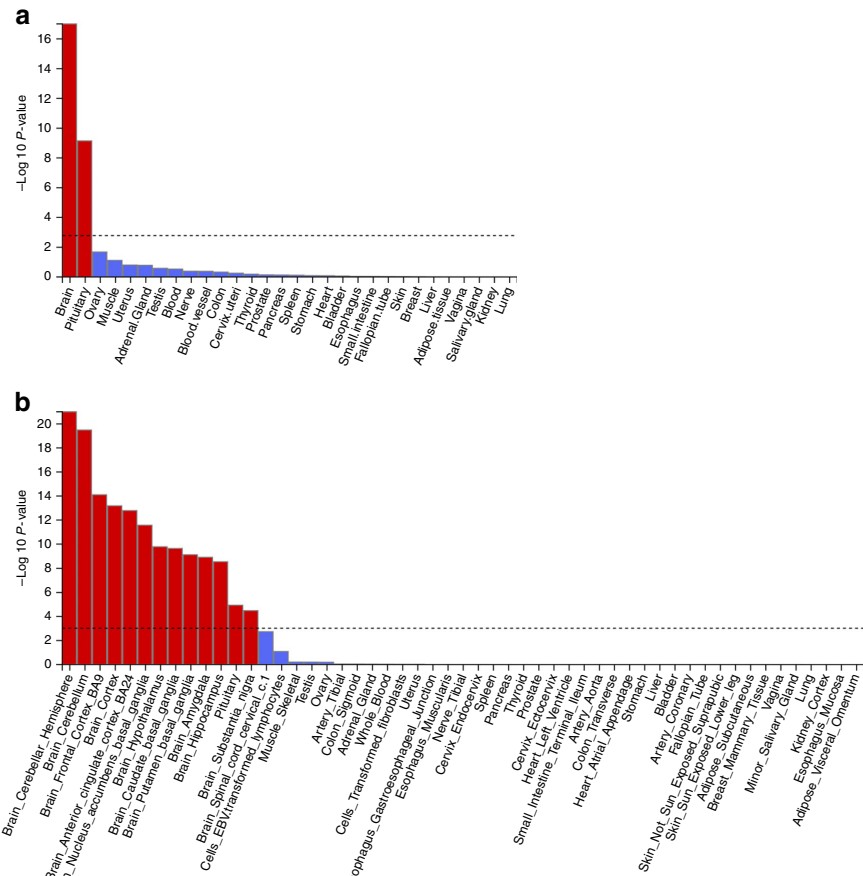

**Fig. 4** MAGMA tissue expression analysis results. Per-tissue enrichment of expression of chronotype genes based on GTEx RNA-seq data for **a** 30 general and **b** 53 specific tissue types

(Supplementary Data 8). The genes in the associated loci were also enriched in multiple pathways relating to the regulation and metabolism of cyclic nucleotides, such as cAMP and cGMP, as well as pathways involved in G-protein signalling and activation. The NMDA glutamate signalling pathway was also enriched and MAGMA-mapped genes in this pathway include *NRXN1* and *RELN*, which have been shown to influence risk of schizophrenia[39,40], but for which there is limited evidence of a role in circadian rhythm regulation.

**Fine-mapping identifies likely causal variants and genes**. To highlight putative causal variants and genes, we fine-mapped the associated loci using FINEMAP[41]. FINEMAP uses a shotgun stochastic search to identify the most plausible causal variant configuration given the GWAS association statistics and local linkage disequilibrium (LD) patterns and outputs the posterior probabilities of each variant configuration being causal. Forty-two loci had a single variant with a probability of >50% of being

causal (Supplementary Data 9). Annotation of these likely causal variants identified ten coding variants. These include a low frequency missense variant in *RGS16* (MAF = 3%, morningness OR = 1.26 for minor allele), previously associated with chronotype[25] and the most strongly associated with morningness in this study, and missense variants in the *INADL, HCRTR2, PLCL1* and *CLN5* genes, all four genes having been identified in previous GWAS[24,26]. Fine-mapping also identified missense variants in *PCYOX1* and *SKOR2*, and a stop gain variant in the *MADD* gene, as likely causal variants in these loci, highlighting further candidate genes for chronotype. The *MADD* stop gain variant (rs35233100) has previously been associated with levels of proinsulin[42], suggesting a potential link between insulin secretion and chronotype. To gain further insight into additional genes that may play a role in determining chronotype, we annotated the putative causal variants using the GTEx eQTL database (Supplementary Data 9). There were 90 variants across 51 loci that were eQTLs for one or more genes, with a total of 208 mapped genes. As an example, this included a putative causal

variant in the promoter of *FBXO3* which represents the strongest eQTL for *FBXO3*. *FBXO3* is in the ubiquitin-proteosome pathway; protein (de)ubiquitination has been shown to be involved with the degradation of several core clock genes[43,44], influencing the build-up up these proteins and the pace of the circadian clock[45,46]. *FBXO3* expression has been shown to be altered by light treatment and to demonstrate rhythmic expression[46].

**SCN-enrichment analysis identifies plausible circadian genes.** The suprachiasmatic nucleus (SCN) is a small region of the brain, consisting of around 20,000 neurons, that is integral to maintaining circadian rhythms in humans and is a likely mechanism of action for at least some of the associated genes and loci. Indeed, the associated loci included many key mammal SCN clock genes including *PER1*, *PER2*, *PER3*, *CRY1*, *FBXL3* and *ARNTL* (Supplementary Data 10). To identify additional genes important in setting and modulating circadian rhythms in the SCN, we assessed expression, enrichment and fluctuation of proximal or eQTL-mapped genes using expression data from the mouse SCN. We cross-referenced all mapped genes at the fine-mapped loci against whether there was evidence for enrichment of expression in the SCN compared to other brain tissue[47,48] and whether the genes demonstrated evidence of fluctuation in expression over the 24-h cycle[47] (Supplementary Data 9). We also annotated the genes against a set of 343 putative clock genes identified from RNAi knockdown experiments a human cellular clock model[49] (Supplementary Data 9). Of the 22.5% of all genes tested that were enriched in the SCN[48], 28.0% of the 804 genes (mapped using MAGMA and present in the enrichment analysis) were enriched in the SCN, representing a significant excess (binomial $P = 2 \times 10^{-4}$). As a negative control, we tested the enrichment of MAGMA-mapped genes for several unrelated GWAS phenotypes, finding no significant excess of SCN-enriched genes (all binomial $P > 0.05$) (Supplementary Table 4). Similar enrichment was found for those chronotype genes fluctuating in the SCN, but no significant excess from the RNAi knockdown study. Enriched and fluctuating genes from the fine-mapping efforts included known circadian genes such as *FBXL3* and putative genes such as *LSM7* and *VIP*. *LSM7* encodes core components of the spliceosomal U6 small nuclear ribonucleoprotein complex for which some previous studies have suggested a role in circadian timing[49,50]. *VIP* encodes a vasoactive peptide hormone that lowers arterial blood pressure and relaxes muscles of the stomach and trachea. Evidence from mouse models indicates that it has a role in generating and light-entrainment of circadian oscillations[51].

**Chronotype is genetically correlated with psychiatric traits.** As a strategy to prioritise traits for subsequent causal analyses, as previous studies have shown a strong correlation between genetic and phenotypic correlations[52,53], and to identify genetic overlap between chronotype and other diseases and traits, we performed LD-score regression analyses against a range of other diseases and traits for which GWAS summary statistics were publicly available (Supplementary Data 11). We estimated the heritability of chronotype to be 13.7% (95% CI: 13.3–14.0%), as calculated by BOLT-REML in the UK Biobank data alone, which is towards the lower end of previously reported figures (12–21%)[24–26]. The most genetically correlated trait was subjective well-being, which was positively correlated with being a morning person ($r_G = 0.17$, $P = 6 \times 10^{-9}$). Psychiatric traits schizophrenia ($r_G = -0.11$, $P = 1 \times 10^{-7}$), depressive symptoms ($r_G = -0.16$; $P = 2 \times 10^{-6}$), major depressive disorder ($r_G = -0.19$; $P = 3 \times 10^{-5}$) and intelligence ($r_G = -0.11$; $P = 8 \times 10^{-6}$) were all negatively correlated with the morning chronotype. Metabolic traits fasting insulin ($r_G = -0.09$, $P = 0.03$) and HOMA-IR ($r_G = -0.12$, $P = 0.009$) were negatively correlated

with being a morning person but did not reach our Bonferroni-corrected significance threshold. Body mass index (BMI) ($r_G = 0.007$, $P = 0.74$) and T2D ($r_G = 0.02$, $P = 0.60$) were not genetically correlated with morningness.

**Evidence of causal link between chronotype and mental health.** Genetic correlations do not allow for statements of causality to be made about the association between an exposure and an outcome. We therefore performed two-sample MR analyses against the five psychiatric traits that showed evidence of a genetic correlation, to estimate causal effects. Because of the extensive literature on the link between chronotype and metabolic disease and because the well-known SNP in *FTO* (rs1558902) previously associated with higher BMI[54,55] was also associated with being a morning person (OR = 1.04, $P = 4.9 \times 10^{-32}$), we also performed two-sample MR against the metabolic phenotypes BMI, type 2 diabetes and fasting insulin levels. For individual instrument effects on chronotype, log ORs (representing liability for morningness) from the secondary morning person meta-analysis were used, as no effect sizes were obtained in the primary meta-analysis. With chronotype as an exposure, we implemented the R package TwoSampleMR[56] to report causal associations of chronotype on these eight outcomes (Supplementary Data 12). We saw evidence that being a morning person confers a liability to lower risk of schizophrenia and greater subjective well-being, with a genetically determined unit log odds increase in self-report morningness being associated with a liability for reduced schizophrenia (OR of 0.89 (0.82–0.96); inverse-variance weighted (IVW) $P = 0.004$) and higher subjective well-being (0.04 SD (0.02–0.06); IVW $P = 5 \times 10^{-5}$), and with good agreement amongst the different MR methods (Figs. 5 and 6). There was suggestive evidence that morningness decreases the liability of depression: one-unit log odds increase in morningness was associated with an OR of 0.65 (0.44–0.95; IVW $P = 0.03$) for major depressive disorder and 0.02 SD lower (0.002–0.04; IVW $P = 0.03$) for depressive symptoms (Supplementary Figures 3 and 4), but these did not reach our multiple testing threshold of $P_{bonf} = 0.005$. There was no strong statistical evidence that chronotype was causally associated with BMI, fasting insulin or risk of type 2 diabetes (IVW $P > 0.1$), as previously reported[24–26].

**No evidence that poor mental health influences chronotype.** To assess whether our genetically correlated phenotypes were causally influencing chronotype, we performed two-sample MR analyses with chronotype as the outcome. Owing to a limited number of genetic instruments, of the original five genetically correlated psychiatric phenotypes we were able to test only schizophrenia and major depressive disorder, in addition to the metabolic phenotypes BMI, insulin secretion and type 2 diabetes (Supplementary Data 13). We observed only weak evidence of liability effects of type 2 diabetes (IVW $P = 0.01$), insulin secretion (IVW $P = 0.04$) and BMI (IVW $P = 0.05$) on chronotype. Despite strong genetic correlations with chronotype, we see no strong evidence that schizophrenia (IVW $P = 0.07$) or major depressive disorder (IVW $P = 0.62$) causally influence liability for morningness.

**Discussion**
Using data from 697,828 individuals, we have performed the largest GWAS study of chronotype and expanded the number of chronotype-associated loci from 24 to 351. Using activity monitor data from 85,760 we showed that these variants are associated with objective measures of sleep timing. We confirm previously reported enrichment of circadian rhythm pathways and retina and brain expressed genes at associated loci, and demonstrate further enrichment of genes in the cAMP, cGMP, NMDA and insulin signalling pathways as well as those in pituitary gland

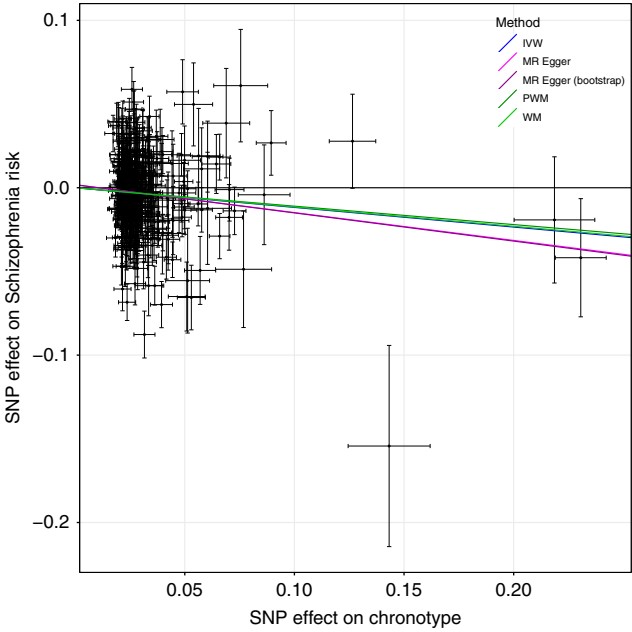

**Fig. 5** MR scatter plot of schizophrenia risk vs. chronotype exposure. Plot shows chronotype meta-analysis variants and their effects (log odds ratios) on schizophrenia risk in the PGC GWAS[77] (outcome) versus odds of being a morning person (exposure). Lines identify the slopes of the five methods tested. Log odds (and SEs) for morningness were taken from the secondary effect-size meta-analysis. Error bars represent standard errors of effect sizes

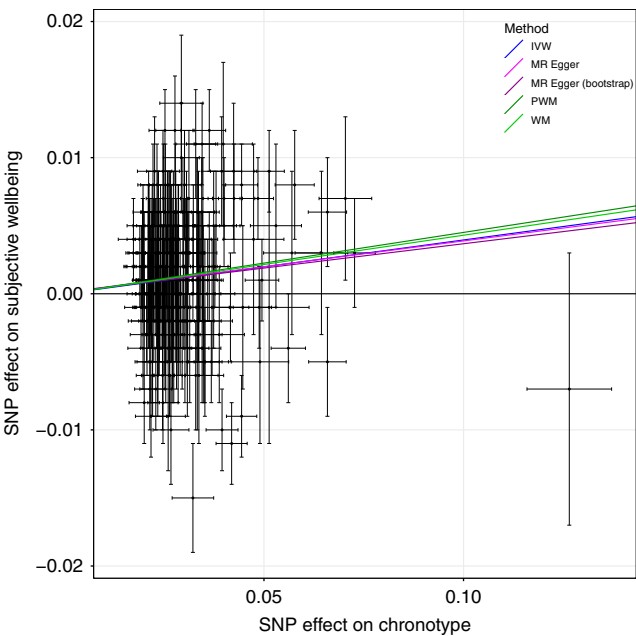

**Fig. 6** MR scatter plot of subjective well-being outcome vs. chronotype exposure. Plot showing chronotype meta-analysis variants and their effects (log odds ratios) on subjective well-being in the SSGAC GWAS[79] (outcome) versus odds of being a morning person (exposure). Lines identify the slopes of the five methods tested. Log odds (and SEs) for morningness were taken from the secondary effect-size meta-analysis. Error bars represent standard errors of effect sizes

tissue and the SCN. We fine-map the loci and provide target genes for other researchers to perform in depth functional investigation into chronobiology. We have provided more accurate genetic correlation estimates of chronotype with a range of

traits and disease and provide some evidence for a causal link between chronotype and mental health.

We have found evidence that the natural variation in circadian preference amongst the human population can be ascribed to several different mechanisms. Given the prominence of genetic variants in or near multiple core circadian rhythm genes (*PER1*, *PER2*, *PER3*, *CRY1*, *FBXL3* and *ARNTL*), we infer that some of the variation is attributed to subtle differences in the biochemical feedback mechanism of the circadian clock. This is supported by evidence of the chronotype-associated loci being enriched in the SCN, suggesting that variants that also subtly affect the modification and regulation of the circadian clock contribute to the population variation of chronotype. Entrainment of circadian rhythms through external stimuli such as light and temperature is well-known but through this study and previous GWAS efforts, we found that an individual's chronotype is also influenced by variants in genes important in the correct formation and functioning of retinal ganglion cells (*RGS16* and *INADL*), highlighting that some natural variation could be explained by better detection and communication (to the SCN) of external light signals. Variants in genes with known roles in appetite regulation (*FTO*), insulin secretion (*MADD*) and even nicotine and caffeine metabolism (*CYP2A6*) point to other processes that impact an individual's chronotype, though it is unclear whether the effect of these on chronotype are mediated through the modulation of the circadian clock or by other means, such as through sleep–wake homoeostasis.

Reported observational associations of chronotype with metabolic diseases are particularly strong[57,58], but we found no evidence for a causal effect of morningness on type 2 diabetes, BMI or insulin levels and could exclude the observational association effect sizes. One possibility which future studies should investigate is whether circadian misalignment, rather than chronotype itself, is more strongly associated with disease outcomes. For example, are individuals who are genetically evening people but have to wake early because of work commitments particularly susceptible to obesity and diabetes?

There are clear epidemiological associations reported in the literature between mental health traits and chronotype, with mental health disorders typically being overrepresented in evening types[59–61], and in this study we show that morningness is negatively genetically correlated with both depression and schizophrenia, and positively correlated with well-being. Previous studies have found a link between schizophrenia and circadian dysregulation and misalignment[62,63] with schizophrenics displaying greater variation in sleep and activity timing and misaligned melatonin and sleep cycles, but no evidence exists for the effect of chronotype on schizophrenia risk. Our MR analyses support a causal role of eveningness on increased risk of schizophrenia, though the statistical significance is not overwhelmingly strong. We do not find evidence of schizophrenia causally influencing chronotype. However, several of the mapped genes at the chronotype-associated loci are well-known schizophrenia loci such as *NRXN1* (as well at *NRXN2* and *NRXN3*) and *RELN*[39,40] and subsequent studies will be necessary to understand the shared biological mechanisms between chronotype and schizophrenia risk.

Chronotype is influenced by circadian rhythms and innate sleep homoeostatic mechanisms, but is also dependent on societal pressures. It is also a self-report measure which means the interpretation of the phenotype and the genetic association is complicated. In this study, however, we show, using objective measures derived from activity monitor data, that these chronotype variants do affect objectives measures of sleep timing, but not other aspects of sleep including duration and timing, providing evidence that we are identifying biologically meaningful associations and allowing us to quantity the effect of these variants on sleep timing.

The response to UK Biobank participation was <5% and this has resulted in selection for healthier individuals, which may introduced bias into our analyses, including in GWAS and MR[64]. Here, GWAS results replicated those of 23andMe, a study that may also suffer from selection bias but of a different nature to UK Biobank. Adopting two-sample MR we attempted to maximise statistical power by using publicly available aggregated data based on consortia of studies that had considerably greater response rates, and avoided winner's curse which can lead to underestimation of causal effects[65]. MR of a binary (or other broad category) exposure that is derived from an underlying continuous trait, as is the case with chronotype, may be biased by horizontal pleiotropy from within-category variation in the trait that cannot be identified by alternative MR methods, such as MR-Egger. As effect sizes for MR analyses were derived from log ORs in the secondary morning person meta-analysis, there may be the possibility of undetected pleiotropy and so our findings should therefore be treated with some caution.

In conclusion, we have identified 327 novel loci that regulate circadian rhythms and sleep timing in humans and provide further insights into the association of chronotype with disease.

## Methods

**Ethics and consent**. The UK Biobank was granted ethical approval by the North West Multi-centre Research Ethics Committee (MREC) to collect and distribute data and samples from the participants (http://www.ukbiobank.ac.uk/ethics/) and covers the work in this study, which was performed under UK Biobank application number 9072. All participants included in these analyses gave informed consent to participate. UK Biobank consent procedures are detailed at http://biobank.ctsu.ox.ac.uk/crystal/field.cgi?id=200. All 23andMe participants were customers of the personal genetics company 23andMe, Inc. and were genotyped for the 23andMe Personal Genome Service. The 23andMe participants included in our analyses provided informed consent for their data to be used for research purposes and responded to online questionnaires according to 23andMe's human subject protocol, which was reviewed and approved by Ethical and Independent Review Services, a private institutional review board (http://www.eandireview.com). Details of 23andMe's consent process can be found at https://www.23andme.com/en-gb/about/consent/.

**Cohorts**. The UK Biobank is a health resource with phenotypic and genetic data on over 500,000 volunteer participants who were aged between 40 and 69. Participants were recruited from the general UK population and baseline data were collected from 2006 to 2010 across 22 centres in England, Scotland and Wales, with recording of detailed anthropometric measures as well as self-report health and sociodemographic variables. The cohort is described in full elsewhere[27]. We used data on 451,454 individuals from the full UK Biobank data release that we identified as White European and that had genetic data available. To define a set of White Europeans, we performed principal components analysis in the 1000 Genomes (1KG) reference panel using a subset of variants that were of a high quality in the UK Biobank. We projected these principal components into the set of related UK Biobank participants to avoid the relatedness confounding the principal components. We then adopted a k-means clustering approach to define a European cluster, initialising the ethnic centres defined by the population-specific means of the first four 1KG principal components. This analysis was performed only within individuals self-reporting as British, Irish, White or Any other white background. Because association analyses are performed using the LMM method, we included related individuals.

We used summary statistics from a morning chronotype GWAS performed by 23andMe of 248,100 participants (120,478 cases, 127,622 controls) with a minimum of 97% European-ancestry. GWAS analysis was performed in a maximal set of unrelated participants, where pairs of individuals were considered related if they shared 700 cM IBD of genomic segments, roughly corresponding to first cousins in an outbred population. The 23andMe cohort is described in more detail elsewhere[24].

**Activity monitor data**. A subset of the UK Biobank cohort was invited to wear a wrist-worn activity monitor for a period of a week. Individuals were mailed the device and asked to wear it continuously for seven days, including while bathing, showering and sleeping. In total, 103,720 participants returned their activity monitor devices with data covering at least three complete 24-hour periods. We downloaded the raw activity monitor data (data-field 90001) for these individuals, in the form of binary Continuous Wave Accelerometer (cwa) files. Further information, along with details of centrally derived variables, is available elsewhere[66]. Detailed protocol information can be found online at http://biobank.ctsu.ox.ac.uk/crystal/docs/PhysicalActivityMonitor.pdf and a sample instruction letter at http://biobank.ctsu.ox.ac.uk/crystal/images/activity_invite.png (UKB Resources 131600 and 141141, respectively; both accessed January 30th 2018). We converted the .cwa files to .wav format using the open-source software omconvert, recommended by the activity monitor manufacturers Axivity,

which is available online (see https://github.com/digitalinteraction/openmovement/tree/master/Software/AX3/omconvert). To process the raw accelerometer data in.wav format, we used the freely available R package GGIR (v1.5-12)[67,68]. The list of our GGIR settings is provided in Supplementary Data 14 and the full list of variables produced by GGIR can be found in the CRAN GGIR reference manual (see https://cran.r-project.org/web/packages/GGIR/GGIR.pdf).

**Genotyping and quality control**. The 23andMe cohort was genotyped on one of four custom arrays: the first two were variants of the Illumina HumanHap550 + BeadChip (4966 cases and 5564 controls), the third a variant of the Illumina OmniExpress + BeadChip (53,747 cases and 61,637 controls) and the fourth a fully custom array (61,765 cases and 60,421 controls). Successive arrays contained substantial overlap with previous chips. These genotypes were imputed to ~15.6 million variants using the September 2013 release of the 1000 Genomes phase 1 reference panel. For analyses, we used ~11.9 million imputed variants with imputation $r^2 \geq 0.3$, MAF $\geq 0.001$ (0.1%) and that showed no sign of batch effects.

The UK Biobank cohort was genotyped on two almost identical arrays. The first ~50,000 samples were genotyped on the UK BiLEVE array and the remaining ~450,000 samples were genotyped on the UK Biobank Axiom array in two groups (interim and full release). A total of 805,426 directly genotyped variants were made available in the full release. These variants were centrally imputed to ~93 M autosomal variants using two reference panels: a combined UK10K and 1000 Genomes panel and the Haplotype Reference Consortium (HRC) panel. For all analyses, we used ~12.0 M HRC imputed variants with an imputation $r^2 \geq 0.3$, MAF $\geq 0.001$ (0.1%) and with a Hardy–Weinberg equilibrium (HWE) $P > 1 \times 10^{-12}$ (chi-squared; 1 degree of freedom). We excluded non-HRC imputed variants on advice from the UK Biobank imputation team. Further details on the UK Biobank genotyping, quality control and imputation procedures can be found elsewhere[28].

**Self-report phenotypes**. Responses to two identical questions ("Are you naturally a night person or a morning person?") were used to define the dichotomous morning person phenotype in the 23andMe cohort, with one question having a wider selection of neutral options. For the first instance, the possible answers were "Night owl", "Early bird" and "Neither", and for the second "Night person", "Morning person", "Neither", "It depends" and "I'm not sure". Individuals with discordant or neutral responses to both were excluded. For those with one neutral and one non-neutral response, their non-neutral response was used to define their phenotype. Morning people were coded as 1 (cases; $N = 120,478$) and evening people were coded as 0 (controls; $N = 127,622$).

The UK Biobank collected a single self-reported measure of Chronotype ("Morning/evening person (chronotype)"; data-field 1180). Participants were prompted to answer the question "Do you consider yourself to be?" with one of six possible answers: "Definitely a 'morning' person", "More a 'morning' than 'evening' person", "More an 'evening' than a 'morning' person", "Definitely an 'evening' person", "Do not know" or "Prefer not to answer", which we coded as 2, 1, −1, −2, 0 and missing, respectively (distribution summarised in Table 1). Of the 451,454 white European participants with genetic data, 449,734 were included in the GWAS (had non-missing phenotype and covariates).

In order to provide interpretable ORs for our genome-wide significant variants, we also defined a binary phenotype using the same data-field as for Chronotype. Participants answering "Definitely an 'evening' person" and "More an 'evening' than a 'morning' person" were coded as 0 (controls) and those answering "Definitely a 'morning' person" and "More a 'morning' than 'evening' person" were coded as 1 (cases). Participants answering "Do not know" or "Prefer not to answer" were coded as missing. A total of 403,195 participants were included in the GWAS (252,287 cases and 150,908 controls).

**Activity monitor phenotypes**. The software package GGIR[68,69] produces quantitative and timing measures relating to both activity levels and sleep patterns, with a day-by-day breakdown, as well averages across the period of wear. A new algorithm, implemented in version 1.5–12 of the GGIR R package and validated using PSG in an external cohort[70], allows for detection of sleep periods without the use of a sleep diary and with minimal bias. Briefly, for each individual, median values of the absolute change in z-angle (representing the dorsal–ventral direction when the wrist is in the anatomical position) across 5-min rolling windows were calculated across a 24-h period, chosen to make the algorithm insensitive to activity monitor orientation. The 10th percentile was incorporated into the threshold to distinguish movement from non-movement. Bouts of inactivity lasting ≥30 min are recorded as inactivity bouts. Inactivity bouts that were <60 min apart were combined to form inactivity blocks. The start and end of longest block defined the start and end of the sleep period time-window (SPT-window).

The UK Biobank made multiple activity monitor data-quality variables available. From our activity monitor phenotypes, we excluded 4925 samples with a non-zero or missing value in data-field 90002 ("Data problem indicator"). We then excluded any individuals with the "good wear time" flag (field 90015) set to 0 (No), "good calibration" flag (field 90016) set to 0 (No), "calibrated on own data" flag (field 90017) set to 0 (No), "data recording errors" (field 90182) > 788 (Q₃ + 1.5 × IQR) or a non-zero count of "interrupted recording periods" (field 90180). Phenotypes determined using the SPT-window (all phenotypes except L5 and M10

timing) had additional exclusions based on short (<3 h) and long (>12 h) mean sleep duration and too low (<5) or too high (>30) mean number of sleep episodes per night (see below). These additional exclusions were to ensure that individuals with extreme (outlying), and likely incorrect, sleep characteristics were not included in any subsequent analyses.

Sleep midpoint was calculated as the time directly between the start and end of the SPT-window and is defined as the number of hours elapsed since midnight at the start of the calendar day on which the STP-window started (e.g., 02:30 = 26.5; 23:45 = 23.75) with a cut-off at midday (12:00 and 36:00). This accounted for participants whose sleep midpoint occurs before midnight. Our sleep midpoint phenotype represented the average of each participant over all their valid SPT-windows. After exclusions and adjustments, 84,810 participants had valid sleep midpoint, covariates and genetic data.

L5 and M10 refer to the least-active five and the most-active 10 h of each day and are commonly studied measures relating to circadian activity and sleep. L5 (M10) defines a 5-h (10-h) daily period of minimum (maximum) activity, as calculated by means of a moving average with a 5-h (10-h) window. As with sleep midpoint, we defined our L5 (M10) timing phenotype as the number of hours elapsed from the previous midnight to the L5 (M10) midpoint, averaged over all valid wear days. Of the 103,711 participants with activity monitor data, there were 85,205 and 85,670 with valid L5 and M10 measures respectively, covariates and genetic data. Basic summaries of these and other raw activity monitor phenotypes are given in Supplementary Table 1.

Sleep episodes within the SPT-window were defined as periods of at least 5 min with no change larger than 5° associated with the $z$-axis of the activity monitor[68]. The summed duration of all sleep episodes provided the sleep duration for a given SPT-window. We took both the mean and standard deviation of sleep duration across all valid SPT-windows to provide a measure of average sleep quantity and a measure of variability. After exclusions and adjustments, we had 85,449 (84,441) participants with valid sleep duration mean (SD), covariates and genetic data.

Sleep efficiency was calculated as a ratio of sleep duration (defined above) to SPT-window duration. The phenotype represented the mean across all valid SPT-windows and after exclusions and adjustments, left us with 84,810 participants with valid sleep efficiency, covariates and genetic data.

The number of sleep episodes was defined as the number of sleep episodes of at least 5 min separated by at least 5 s of wakefulness within the SPT-window. The phenotype represented the mean across all SPT-windows and, once adjusted for the mean length of time in bed, can be interpreted as a measure of sleep disturbance or fragmentation. After exclusions and adjustments, we had 84,810 participants with a valid number of sleep episodes, covariates and genetic data.

Diurnal inactivity was defined as the total daily duration of estimated bouts of inactivity that fall outside of the SPT-window. This comprised the total length of periods of sustained inactivity (>5 min) and captured sleep (naps), but did not include other inactivity such as sitting and reading or watching television, which involve a low but detectable level of movement. This variable likely captured some non-sleep rest as it was not possible to separate these without detailed activity diaries. The phenotype was calculated as the mean across all valid days and, after exclusions and adjustments, we were left with 84,757 participants with a valid measure, covariates and genetic data.

**Genome-wide association analysis**. We performed all association test using BOLT-LMM[71] v2.3, which applies a linear mixed model (LMM) to adjust for the effects of population structure and individual relatedness, and allowed us to include all related individuals in our white European subset, boosting our power to detect associations. This meant a sample size of up to 449,734 individuals, as opposed to the set of 379,768 unrelated individuals. BOLT-LMM approximates relatedness within a cohort by using LD blocks and avoids the requirement of building a genetic-relationship matrix, with which calculations are intractable in cohorts of this size. From the ~805,000 directly genotyped (non-imputed) variants available, we identified 524,307 high-quality variants (bi-allelic SNPs; MAF ≥ 1%; HWE $P > 1 \times 10^{-6}$; non-missing in all genotype batches, total missingness < 1.5% and not in a region of long-range LD[72]) that BOLT-LMM used to build its relatedness model. For LD structure information, we used the default 1000 Genomes LD-Score table provided with the software. We forced BOLT-LMM to apply a non-infinitesimal model, which provides better effect size estimates for variants with moderate to large effect sizes, in exchange for increased computing time. At runtime, the chronotype and morning person phenotypes were adjusted for age (field 21003), sex (field 31), study centre (field 54; categorical) and a derived variable representing genotyping release (categorical; UKBiLEVE array, UKB Axiom array interim release and UKB Axiom array full release). Accelerometer-based phenotypes were adjusted for age activity monitor worn (derived from month and year of birth and date activity monitor worn), sex, season activity monitor worn (categorical; winter, spring, summer or autumn; derived from date activity monitor worn) and number of valid measurements (SPT-windows for sleep phenotypes, number of valid days for diurnal inactivity or number of L5 or M10 detections for L5 or M10 timing). The GWA analysis for the number of sleep episodes phenotype was also adjusted for the mean length of SPT-window (across all included SPT-windows) to account for the fact that individuals have a greater number of sleep episodes the longer they spend in bed.

In the 23andMe morning person GWAS, the summary statistics were generated through logistic regression (using an additive model) of the phenotype against the genotype, adjusting for age, sex, the first four principal components and a

categorical variable representing genotyping platform. Genotyping batches in which particular variants failed to meet minimum quality control were not included in association testing for those variants, resulting in a range of sample sizes over the whole set of results. A $\lambda_{GC}$ of 1.325 was reported for this GWAS. Lead variants for the 23andMe only morning person GWAS are provided in Supplementary Data 15.

**Sensitivity analysis**. To avoid issues with stratification, we performed a sensitivity GWAS, in UK Biobank alone, to assess whether any of the associations were driven by a subset of the cohort with specific conditions. We excluded those reporting shift or night shift work at baseline, those taking medication for sleep or psychiatric disorders and those with either with a HES ICD10 or self-reported diagnosis of depression, schizophrenia, bipolar disorder, anxiety disorders or mood disorder (see Supplementary Methods for further details). Results for the 341 lead chronotype variants available in the UK Biobank are provided in Supplementary Data 1 alongside the main meta-analysis results.

**Meta-analysis of GWAS results**. Meta-analysis was performed using the software package METAL[73]. To obtain the largest possible sample size, and thus maximising statistical power, we performed a sample-size meta-analysis, using the results from the UK Biobank chronotype GWAS and the 23andMe morning person GWAS. Genomic control was not performed on each set of summary statistics prior to meta-analysis but instead the meta-analysis chi-squared statistics were corrected using the LD-score intercept ($I_{LDSC} = 1.0829$), calculated by the software LDSC, as using $\lambda_{GC}$ is considered overly conservative and the LD-score intercept better captures inflation due to population stratification[74]. For interpretable results, we reported the OR from a secondary effect size meta-analysis between our dichotomous UK Biobank morning person GWAS and the 23andMe morning person GWAS. The primary chronotype sample-size meta-analysis produced results for 15,880,941 variants in up to 697,828 individuals, with the secondary effect-size morning person meta-analysis producing results for 15,880,664 variants in up to 651,295 individuals (372,765 cases and 278,530 controls). All reported meta-analysis $P$ values were calculated by METAL using a $Z$-test.

**Post-GWAS analyses**. We used MAGENTA[37], DEPICT[38], PASCAL[36] and MAGMA[34] to perform pathway and tissue enrichment. For MAGENTA and DEPICT, we included all variants from the meta-analysis, whereas for PASCAL, we included only those with an RSID as the software assigns variants to genes using their RSID. For the MAGENTA analysis, we used upstream and downstream limits of 110Kb and 40Kb to assign variants to genes by position, we excluded the HLA region from the analysis and set the number of permutations for gene-set enrichment analysis to 10,000. For DEPICT, we used the default settings and the annotation and mapping files provided with the software. As each of the four pieces of software adopts a different gene prioritisation method or relies on different databases, we included results from all four to cover all bases and to allow for better comparison with other studies, where only a single method may have been used. Briefly PASCAL corrects for the effect of LD blocks by accounting for the LD structure between associated variants, MAGENTA uses distance-based mapping but allows the user to set the upstream and downstream distances for inclusion, DEPICT makes use of large-scale data on gene co-regulation to prioritise genes before calculating enrichment in its own reconstituted gene sets and MAGMA, the most recent method (and implemented in the FUMA GWAS[35] platform), claims greater statistical power to detect enriched gene sets than methods such as MAGENTA and PASCAL, without affecting the type 1 error rate. By using multiple methods and looking for consistency, we provide more compelling evidence of enrichment in specific pathways and tissues.

We used the LD-score regression (LDSC) software, available at https://github.com/bulik/ldsc/, to quantify the genetic overlap between the trait of interest and 222 traits with publicly available GWA data. Briefly, to estimate heritability of a single phenotype, LDSC regresses chi-squared statistics from summary statistics against pre-computed LD Scores (a measure of how well each variant tags nearby variants) for all variants of the phenotype. The genetic correlation ($r_g$) between two phenotypes is, similarly, calculated by regressing each variant's product of $Z$-scores from the two phenotypes against the LD scores; the slope of the regression line is the estimate of $r_g$. The $P$ values reported in this manuscript were calculated using a $Z$-test of calculated $r_g$ against the null hypothesis of $r_g = 0$. Further methodological details are given elsewhere[74]. We used an LD-Score panel calculated in European samples from 1000 Genomes phase 3 v5 and removed variants that were not present in this reference panel. We considered any correlation as statistically significant if it had a Bonferroni-corrected $P < 0.05$.

Fine-mapping analyses were performed using FINEMAP v1.1[41] using a shotgun stochastic approach, allowing up to 20 causal SNPs at each locus and by focussing on a 1 Mb (±500 Kb) region around each index variant. As FINEMAP assumes a fixed sample size for all variants, we excluded variants not present in both the UK Biobank and 23andMe data, and to make the LD calculations more tractable we excluded variants with GWA analysis $P > 0.01$ to limit the total number of variants at each locus. We constructed an LD matrix for each locus by calculating the Pearson correlation coefficient for all pairs of variants using dosages derived from the unrelated European-ancestry subset of the UK Biobank imputed genotype probabilities ($N = 379,769$). A variant was considered to be causal if its $\log_{10}$ Bayes

factor was 2 or larger, a limit recommended by the FINEMAP documentation (http://www.christianbenner.com/index_v1.1.html).

We annotated variants identified by FINEMAP as likely to be causal using Alamut Batch v1.8 (Interactive Biosoftware, Rouen, France) with genome assembly GRCh37 and all options set to default. We retained only the canonical (longest) transcript for each variant and reported the variant location and coding effect (if applicable) in this transcript. To identify whether variants were cis-eQTLs for nearby genes, we performed a lookup of our variants in the GTEx single-tissue cis-eQTL dataset (v7), accessed at the GTEx portal (https://www.gtexportal.org/home/datasets) on 13/07/18, for significant associations. A variant was reported as an eQTL for a gene if the variant-gene association was significant ($q$ value ≤ 0.05) for one or more brain or non-brain tissues.

With the aim of highlighting genes that have a role in regulating the internal circadian clock, we cross-referenced the genes identified by eQTL mapping, in addition to the two nearest genes (within 1 Mb), with catalogues from three gene expression studies. Firstly, we used data from an RNAi screen of circadian clock modifiers[49], in which a genome-wide scan was performed on the effects of single-gene knockouts on the amplitude and period of the circadian expression. Secondly, we used data from a study of gene expression in SCN tissue over a 24-h light/dark cycle[47] to identify whether our genes exhibit fluctuating expression in SCN tissue and whether the genes show enriched expression in the SCN compared to other tissues. Finally, we used data from a meta-analysis of gene expression in the SCN[48] to investigate whether the genes were preferentially expressed in the SCN when compared to other brain tissues.

**MR analyses**. We undertook MR analyses to explore both the effect of chronotype on different outcomes and the effect of different exposures on chronotype as an outcome. These two-sample MR analyses can be summarised by:

1. Chronotype exposure using the 351 variants and effect sizes discovered in this meta-analysis against the five significant psychiatric outcomes from the genetic correlation analyses and three metabolic outcomes, using summary data from published GWAS (Supplementary Data 12).
2. Two of the five significant psychiatric exposures from the genetic correlation analyses and four metabolic exposures, all using variants from published GWAS, against chronotype as an outcome, using summary data from this meta-analysis (Supplementary Data 13).

In both analyses, we tested four MR methods:

a. Inverse-variance weighting (IVW)[75]
b. MR-Egger[75]
c. Weighted median (WM)[76]
d. Penalised weighted median (PWM)[76]

Analysis 1 (chronotype exposure) was performed using the R package TwoSampleMR using aggregated summary statistics available through the MR-Base platform[56]. We implemented the four MR methods listed above and also included the MR-Egger bootstrap to provide better estimates of the effect sizes and standard errors as compared to the MR-Egger method. We used data from published GWAS to test the effect of chronotype on the following exposures: schizophrenia[77], major depressive disorder[78], depressive symptoms[79], subjective well-being[79], PGC cross-disorder traits[80], fasting insulin[81], BMI[55,82] and T2D[83,84]. To provide meaningful effect sizes for MR analyses, we used betas from the secondary effect size meta-analysis of the dichotomous UK Biobank and 23andMe morning person GWAS.

For analysis 2 (chronotype outcome) we applied the four MR methods listed above, utilising a custom pipeline. Using data from published GWAS, we tested whether chronotype is influenced by the following exposures: schizophrenia[77], major depressive disorder[78,85], insulin secretion[86], favourable adiposity[87], BMI[55] and T2D[88]. As with analysis 1, chronotype effect sizes represented morningness liability and were taken from the secondary morning person meta-analysis, with the exception of the major depressive disorder exposure from a 23andMe study[85] for which outcome effect sizes were taken from the UK Biobank-only chronotype GWAS.

We used the inverse-variance weighted approach as our main analysis method and MR-Egger, weighted median estimation and penalised weighted median estimation as sensitivity analyses in the event of unidentified pleiotropy of our genetic instruments. MR results may be biased by horizontal pleiotropy, i.e., where the genetic variants that are robustly related to the exposure of interest (here chronotype) independently influence the outcome, through association with another risk factor for the outcome. IVW assumes that there is either no horizontal pleiotropy (under a fixed effect model) or, if implemented under a random effects model after detecting heterogeneity amongst the causal estimates, that:

I. The strength of association of the genetic instruments with the risk factor is not correlated with the magnitude of the pleiotropic effects.
II. The pleiotropic effects have an average value of zero.

MR-Egger provides unbiased causal estimates if just the first condition above holds, by estimating and adjusting for non-zero mean pleiotropy. However, MR-Egger requires that the InSIDE (Instrument Strength Independent of Direct Effect) assumption[89] holds, in that it needs the pleiotropy of the genetic instruments to be uncorrelated with the instruments' effect on the exposure. The weighted median approach is valid if less than 50% of the weight in the analysis stems from variants that are pleiotropic (i.e., no single SNP that contributes 50% of the weight or a number of SNPs that together contribute 50% should be invalid because of horizontal pleiotropy). Given these different assumptions, if all methods are broadly consistent this strengthens our causal inference. IVW causal effect size estimate $P$ values were calculated using Student's $t$ test with ($N_{SNP}$-1) degrees of freedom, MR Egger using Student's $t$ test with ($N_{SNP}$-2) degrees of freedom and WM/PWM using a Z-test. Additional care should be taken interpreting results from binary exposures or outcomes, as these MR methods assume that horizontal pleiotropy due to within-category variation of dichotomous or categorical traits is negligible.

## Data availability

Summary statistics for the top 10,000 chronotype meta-analysis variants are provided in Supplementary Data 10. The full set of UK Biobank-only chronotype and morning person GWAS summary statistics can be found at http://www.t2diabetesgenes.org/data/ and on the Sleep Disorder Knowledge Portal at http://sleepdisordergenetics.org/informational/data/. Full meta-analysis summary statistics can be requested directly from 23andMe Inc. (see https://research.23andme.com/collaborate/#publication). The GGIR R script used to generate the activity monitor measures (Supplementary Data 14) is available with the online version of this article.

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

## Acknowledgements

This research has been conducted using the UK Biobank Resource (application 9072). We would like to thank the research participants and employees of 23andMe for making this work possible. We also wish to acknowledge the Genotype-Tissue Expression (GTEx) Project, supported by the Common Fund of the Office of the Director of the National Institutes of Health, and by NCI, NHGRI, NHLBI, NIDA, NIMH and NINDS, for providing access to the data we required. S.E.J. is funded by the Medical Research Council (Grant MR/M005070/1) M.A.T., M.N.W., and A.M. are supported by the Wellcome Trust Institutional Strategic Support Award (WT097835MF). A.R.W., T.M.F., and H.Y. are supported by the European Research Council grants: SZ-245 50371-GLUCOSEGENES-FP7-IDEAS-ERC and 323195. R.M.F. is a Sir Henry Dale Fellow (Wellcome Trust and Royal Society grant: 104150/Z/14/Z). R.N.B. is funded by the Wellcome Trust and Royal Society grant: 104150/Z/14/Z. J.T. is funded by a Diabetes Research and Wellness Foundation Fellowship. J.B. and D.A.L. work in a Unit that receives support from the University of Bristol and UK Medical Research Council (MC_UU_00011/3 and MC_UU_00011/6, respectively). D.A.L.'s contribution to this work is also supported by the European Research Council under the European Union's Seventh Framework Programme (FP/2007-2013)/ERC Grant Agreement (Grant number 669545; DevelopObese). D.W.R. is supported by Wellcome Investigator award 107849/Z/15/Z. J.M.L. is supported by NIH grants F32DK102323 and 4T32HL007901. M.K.R. is supported by The University of Manchester Research Infrastructure Fund. E.M.B. is supported by grants from the National Health and Medical Research Council of Australia 1145645, 1078901 and 1087889. H.S.D. and R.S. are supported by NIH R01DK107859, NIH R01DK102696 and the MGH Research Scholar Fund. H.T. is supported by Dutch Medical Research Foundation grants (016.VICI.170.200 and VIDI 017.106.370). The funders had no influence on study design, data collection and analysis, decision to publish or preparation of the manuscript.

## Author contributions

S.E.J., J.M.L., A.R.W., H.S.D., D.A.L., M.K.R., R.S. and M.N.W. contributed to the design of the study, S.E.J., J.M.L., A.R.W., V.T.v.H., J.T., R.N.B., A.R.J., H.S.D., K.S.R., M.A.T., H.Y., S.A.S., Y.J., W.D.T., J.W.H., J.B., D.R.M., P.R.G., 2.R.T., D.A.L., T.M.F., M.K.R., D. A.H. and M.N.W. acquired, analysed and/or interpreted the data, S.E.J., J.M.L., A.R.W., V.T.v.H., J.T., R.N.B., A.R.J., H.S.D., M.H., A.D., E.M.B., H.T., K.V.A., D.W.R., R.M.F., A.M., D.R.M., P.R.G., D.A.L., T.M.F., R.S. and M.N.W. drafted and/or made important contributions to the article and V.T.v.H., J.B., P.R.G., T.M.F. and M.K.R. either provided technical or supervisory support for this work.

## Additional information

## Additional information

**Competing interests:** D.A.L. receives support from Roche Diagnostics and Medtronic for research that is unrelated to this study. M.K.R. reports receiving honoraria and consulting fees from Novo Nordisk, Ascensia, Cell Catapult and Roche Diabetes Care. K.V.A. received support from SANOFI-Aventis for research that is unrelated to this study. D.A.H. and 2.R.T. are employees of, and hold stock or stock options in, 23andMe Inc. The remaining authors declare no competing interests.

## 23andMe Research Team

Michelle Agee[18], Babak Alipanahi[18], Adam Auton[18], Robert K. Bell[18], Katarzyna Bryc[18], Sarah L. Elson[18], Pierre Fontanillas[18], Nicholas A. Furlotte[18], Karen E. Huber[18], Aaron Kleinman[18], Nadia K. Litterman[18], Jennifer C. McCreight[18], Matthew H. McIntyre[18], Joanna L. Mountain[18], Elizabeth S. Noblin[18], Carrie A.M. Northover[18], Steven J. Pitts[18], J. Fah Sathirapongsasuti[18], Olga V. Sazonova[18], Janie F. Shelton[18], Suyash Shringarpure[18], Chao Tian[18], Joyce Y. Tung[18], Vladimir Vacic[18] & Catherine H. Wilson[18]

