## [Peer Review File · Nature Communications]

Reviewer #1 (Remarks to the Author):

Although this MS is much improved, I still cannot fully appreciate its contribution as a worthwhile publication. Many previous concerns were addressed, but not in full. The lack of sharing of full results also remains a concern (and also even limits the paper itself, because in the reply letter the authors state “Estimates of heritability for 23andMe have not been included as we do not have individual

level data”: one would hope / assume that at least some of the authors to have access to these data?).

There are multiple instances of nearly incomprehensible expressions (e.g. “Genetic correlations do not allow for magnitudes of causality to be determined ...”: what are magnitudes of causality? Or in the abstract: “the mean sleep timing of the 5% of individuals 55 carrying the most “morningness” alleles was 25 minutes earlier”: what is sleep timing? When people get up? Or when they go bed? Or something else?).

In the way of presenting why an analysis was done, a rationale such as “Because of the variety of methods available and databases employed....” is not sufficient: indicate what different goals the variety might serve.

It is obvious that a relationship between genetic and phenotypic correlations must exist for heritable phenotypes, but why this should be a reason to estimate genetic correlations is not very clear. The heading “Chronotype is heritable and demonstrates strong genetic correlation with several psychiatric traits” seems misleading: correlations do not exceed 0.17 (which is not very strong?). To obtain a phenotypic correlation, the genetic correlation should be weighted by the square root of the heritabilities of the 2 traits, but we do not know the heritability of the chronotype as the only estimate that is provided is by BOLT-REML in the UK Biobank data alone (assuming that the genetic correlation was based on the entire combined 23andme dataset plus UKB).

Reviewer #2 (Remarks to the Author):

The manuscript is substantially improved in its revised form for this journal.

My only residual comment regards the choice of data for MR when modelling chronotype as the outcome. The genetic risk score for depression must be very limited based on the paper cited, and there are far bigger analyses currently available. One solution would be to use the variants discovered by 23andMe (<https://www.ncbi.nlm.nih.gov/pubmed/27479909>) to predict into the UKBB only component of the study. Some other more powerful permutation may also be available based on the other recently published meta-analyses. It seems somewhat unlikely that major psychiatric disease wouldn't impact sleep duration/quality, so every effort should be made to use the best genetic score to test this.

Reviewer comments:

Reviewer #1:

Although this MS is much improved, I still cannot fully appreciate its contribution as a worthwhile publication. Many previous concerns were addressed, but not in full. The lack of sharing of full results also remains a concern (and also even limits the paper itself, because in the reply letter the authors state “Estimates of heritability for 23andMe have not been included as we do not have individual

level data”: one would hope / assume that at least some of the authors to have access to these data?).

Response: We thank the reviewer for their opinion but we are disappointed that the reviewer feels that this study does not contribute much to the literature. Although we replicate some of the findings from our previous paper such as some of the genomic loci and pathway enrichment, we (amongst other things) demonstrate new findings such as 327 additional loci (up from 24) that tell us more about the mechanisms influencing chronotype, find the likely causal variants in these loci through fine-mapping and show the effect of morningness genetics on real activity and sleep timing from the largest objective sleep dataset ever collected.

We agree with the reviewer that it is unfortunate that the policy of 23andMe is only to share the summary data for the most significant 10,000 variants. This policy applies to all approved sharing of 23andMe data (for any project) and is done to ensure that no individual participant can be identified; it is central to their ethical and governance procedures and agreements with participants. Our findings and the results that we are able to share have benefitted greatly from the inclusion of data from the 23andMe cohort. For instance, we had more power to detect association signals which serve as genetic instruments in Mendelian Randomisation analyses. We have already provided clear methodological details of the fact that we are using summary data from 23andMe and an availability statement (before the Acknowledgements section) with details on how to request summary statistics from 23andMe. It is not necessary for authors of a paper to have individual level participant data for all studies to make an important contribution to the literature, as is demonstrated by the numerous large scale meta-analyses that have had major impact on clinical policy and the widespread use of ‘in silico’ data in the genomics and other omics fields. What is important is that data access and use are fully and clearly reported as we have done here.

There are multiple instances of nearly incomprehensible expressions (e.g. “Genetic correlations do not allow for magnitudes of causality to be determined ...”: what are magnitudes of causality? Or in the abstract: “the mean sleep timing of the 5% of individuals 55 carrying the most “morningness” alleles was 25 minutes earlier”: what is sleep timing? When people get up? Or when they go bed? Or something else?).

Response: We agree with the reviewer that the “magnitudes of causality” sentence may have been unclear, so we have revised it for clarity. In terms of phrasing in the abstract, we believe that mean sleep timing is clear; in the main body of the results, we explain that this difference is determined using minimum activity (L5) timing.

In the way of presenting why an analysis was done, a rationale such as “Because of the variety of

methods available and databases employed....” is not sufficient: indicate what different goals the variety might serve.

Response: We had already explained the differences between software packages in the Methods section (see “Pathway analysis and tissue-enrichment” subsection) but for clarification, we have provided more justification in the results section – the sentence now starts “Because of the variety of methods available and databases employed, and to provide a better comparison with studies that have implemented other methods, ...”.

It is obvious that a relationship between genetic and phenotypic correlations must exist for heritable phenotypes, but why this should be a reason to estimate genetic correlations is not very clear. The heading “Chronotype is heritable and demonstrates strong genetic correlation with several psychiatric traits” seems misleading: correlations do not exceed 0.17 (which is not very strong?). To obtain a phenotypic correlation, the genetic correlation should be weighted by the square root of the heritabilities of the 2 traits, but we do not know the heritability of the chronotype as the only estimate that is provided is by BOLT-REML in the UK Biobank data alone (assuming that the genetic correlation was based on the entire combined 23andme dataset plus UKB).

Response: We agree with the reviewer in that we have not necessarily made it clear why we have relied on genetic correlations to guide our MR analyses. As genetic correlations can be derived using just the publicly-available summary statistics, we can calculate genetic correlations (and infer phenotypic correlations) for phenotypes that are a) under-represented, b) not recorded and c) not well-defined in our dataset (i.e. the UK Biobank). To remain consistent, we used genetic correlations to guide MR analyses instead of a combination of genetic and phenotypic correlations. Further justification is that we are interested in assessing causality of phenotypes where there is some evidence of genetic overlap, as these are the phenotypes for which we will have statistical power to assess direction of association through Mendelian Randomisation.

In terms of the wording “strong correlations”, strong refers to the strength of significance and not the magnitude of correlation. To avoid any ambiguity, we have changed the word “strong” to “statistically significant” in the title of that section.

With regards to the heritability of chronotype, the estimate provided by BOLT-REML is indeed calculated using only the UK Biobank, as the best methods currently require individual-level data, but there is no reason to believe that this estimate would be unrepresentative in relation to a similar cohort such as 23andMe. As we are using genetic correlation to identify overlapping phenotypes, there is no reason to estimate phenotypic correlation using the heritability estimate and genetic correlations.

Reviewer #2:

The manuscript is substantially improved in its revised form for this journal.

My only residual comment regards the choice of data for MR when modelling chronotype as the outcome. The genetic risk score for depression must be very limited based on the paper cited, and there are far bigger analyses currently available. One solution would be to use the variants discovered by 23andMe (<https://www.ncbi.nlm.nih.gov/pubmed/27479909>) to predict into the UKBB only component of the study. Some other more powerful permutation may also be available based on the other recently published meta-analyses. It seems somewhat unlikely that major

psychiatric disease wouldn't impact sleep duration/quality, so every effort should be made to use the best genetic score to test this.

Response: We thank the reviewer for their feedback and acknowledge that we did not use the latest study for testing major depressive disorder (MDD) as an exposure. We used the reported MDD-associated variants from the publication highlighted by the reviewer (<https://doi.org/10.1038/ng.3623>) and as this study was performed using 23andMe data, we assessed the effect of MDD as an exposure on chronotype using the UKB-only summary statistics. We still find no evidence to support the hypothesis that MDD influences chronotype (IVW $P=0.97$); we have included these result in Supplementary Table 17.

Reviewer #2 (Remarks to the Author):

I am satisfied with the author response to my residual concern (just a shame it wasn't significant!).

For what it's worth, I also think the responses to review 1 are highly appropriate and do not warrant further work. It is quite unfair to suggest this manuscript does not represent a worthwhile publication.